# A Cross-Sectional Survey on the Clinical Management of Emergence Delirium in Adults: Knowledge, Attitudes, and Practice in Mainland China

**DOI:** 10.3390/brainsci12080989

**Published:** 2022-07-26

**Authors:** Yi Yuan, Bao Lei, Zhengqian Li, Xiaoxiao Wang, Huiling Zhao, Meng Gao, Yingying Xue, Wenchao Zhang, Rui Xiao, Xue Meng, Hongcai Zheng, Jing Zhang, Geng Wang, Xiangyang Guo

**Affiliations:** 1Department of Anesthesiology, Beijing Jishuitan Hospital, No. 31, Xinjiekou East Street, Xicheng District, Beijing 100035, China; julietyy@sina.com (Y.Y.); wenchaozhang116@163.com (W.Z.); xiaorui_beiyi@sina.com (R.X.); mmmxxx888666@126.com (X.M.); 2Department of Anesthesiology, The Yan’an Branch of Peking University Third Hospital, Yan’an Traditional Chinese Medicine Hospital, Yan’an 716000, China; leibaobysy@126.com (B.L.); lizhengqianmazui@126.com (Z.L.); zhlyingxi@163.com (H.Z.); meg_99@126.com (M.G.); sunshun2022@126.com (Y.X.); 3Department of Anesthesiology, Peking University Third Hospital, No. 49, North Garden Street, Haidian District, Beijing 100191, China; 4Perioperative Medicine Branch of China International Exchange and Promotive Association for Medical and Health Care (CPAM), No. 49, North Garden Street, Haidian District, Beijing 100191, China; tongtong030806@163.com (H.Z.); prettyjingzh@126.com (J.Z.); 5Research Center of Clinical Epidemiology, Peking University Third Hospital, No. 49, North Garden Street, Haidian District, Beijing 100191, China; 13051291428@163.com

**Keywords:** emergence delirium, PACU, anesthesia nurse

## Abstract

Background: Emergence delirium (ED) occurs immediately after emergence from general anesthesia, which may have adverse consequences. This cross-sectional survey assessed Chinese physicians’ and nurses’ knowledge of, attitudes towards, and practice regarding ED in adults. Methods: Electronic questionnaires were sent to 93 major academic hospitals across mainland China and both attending anesthesiologists and anesthesia nurses were recommended to complete them. Results: A total of 243 anesthesiologists and 213 anesthesia nurses participated in the survey. Most of the participants considered it a very important issue; however, less than one-third of them routinely assessed ED. In terms of screening tools, anesthesiologists preferred the Confusion Assessment Method, while anesthesia nurses reported using multiple screening tools. Divergence also appeared with regard to the necessity of monitoring the depth of anesthesia. Anesthesiologists considered it only necessary in high-risk patients, while the nurses considered that it should be carried out routinely. No unified treatment strategy nor medication was reported for ED treatment during the recovery period. Conclusions: This study illustrated that there are high awareness levels among both Chinese anesthesiologists and anesthesia nurses regarding the importance of ED. However, a specific practice in terms of routine delirium assessment, anesthesia depth monitoring, and a standardized treatment algorithm needs to be implemented to improve ED management.

## 1. Introduction

Emergence delirium (ED) refers to the very early onset of postoperative delirium (POD) in the immediate post-anesthesia period, before or on arrival to the recovery room [1]. Such delirium may become dangerous and lead to several adverse consequences, including injury, hemorrhage, self-extubation, removal of catheters, or prolonged cognitive dysfunction [2,3]. According to the European Society of Anesthesiology (ESA), in its evidence- and consensus-based guideline for the prevention and treatment of POD, ED can present in hyperactive (referred to as emergence agitation), hypoactive (referred to as hypoactive emergence), or mixed forms [4]. Generally, hypoactive emergence is more common in older adult patients and can easily be overlooked, thereby delaying timely treatment [1]. The incidence of ED (excluding psychomotor subtypes) is approximately 3–21.1% [5,6,7]. Despite the importance of early identification and prompt management of delirium, ED in the post-anesthesia care unit (PACU) has not been comprehensively investigated, especially in adults and older adult patients.

There have been two previous investigations on the current practice of delirium assessment and management among Chinese physicians. The data from anesthesiologists showed a high level of awareness regarding the importance of POD but a relatively low actual practice rate, including routine assessment and monitoring, while the data reported for physicians in intensive care units showed that only half of them screened for delirium daily using an assessment scale [8,9]. Therefore, a gap between clinical attitude and practice was found to exist. Nevertheless, whether the same gap exists for ED is unknown. Since the ESA guideline indicated that nurses in the PACU and ICU play an important role in early recognition of delirium [1], both physicians’ and nurses’ knowledge levels and practice need to be investigated. Furthermore, no previous comparative data regarding the current status of practice for either POD or ED among physicians and nurses has ever been reported. Therefore, the purpose of this current survey was to investigate the clinical management of ED in adults, including knowledge, attitudes, and practice, among both anesthesiologists and anesthesia nurses in mainland China.

## 2. Materials and Methods

### 2.1. Survey Design and Target Population

The authors designed the current cross-sectional survey. An electronic questionnaire was developed in Chinese by the authors and reviewed by the Perioperative Medicine Branch of the China International Exchange and Promotive Association for Medical and Health Care. The committee approved this as a national survey. The institutional ethical review board of Beijing Jishuitan Hospital, the institution of the first author (YY), also approved this survey targeting anesthesiologists and anesthesia nurses across mainland China (JLK202109-53).

The questionnaire consisted of 14 questions that were subdivided into six sections: general demographic data of respondents, the importance of ED, risk factors and prevention measures, assessment of ED, monitoring of the depth of anesthesia (DOA), and treatment for ED. The survey was proofread and then launched via WeChat, a mobile software application invented by Tencent Mobile International Limited (please refer to the Appendix A for the completed version of the questionnaire, Appendix A). The survey was accessible through any web-enabled mobile device available to run the WeChat application.

### 2.2. Data Sampling

The survey was conducted from 1 to 15 October 2020. The electronic questionnaire was sent to 93 major academic teaching hospitals in the mainland of China. We contacted local representatives (committee members) of the perioperative medicine subcommittee in 93 major academic teaching hospitals in mainland China, three in each provincial capital of all 31 provinces and municipalities. Each committee member was asked to complete the survey via the WeChat application, and we invited the anesthesia head nurse to complete the same questionnaire, too. All participants in the study provided electronic informed consent.

### 2.3. Statistical Analysis

Descriptive statistics were calculated. Fisher’s exact *t*-test was used for physician and nurse comparisons. All statistical analysis was performed using IBM SPSS version 25 (IBM Corp., Armonk, NY, USA); *p* < 0.05 was considered statistically significant. Figures were created using GraphPad Prism version 8.0.0 for windows (GraphPad Software, San Diego, CA, USA). 

## 3. Results

### 3.1. General Data of Respondents

The participants were from 3 hospitals of each provincial capital of all 31 provinces and municipalities across China, that is, 93 hospitals in total. We included three anesthesiologists and three anesthesia nurses from each hospital. To be more specific, the anesthesiologists we chose were: one resident in training, one attending doctor, and one associate chief physician. The anesthesia nurses we chose were: one nurse, one supervisor nurse, and one co-chief superintendent nurse. Altogether, 243 anesthesiologists and 213 anesthesia nurses completed the survey, giving a response rate of 87.1% among anesthesiologists and 76.3% among anesthesia nurses.

In terms of work experience, 54.7% (*n* = 133) of anesthesiologists had more than 10 years of experience in their field, 81 (33.3%) had been working for 5–10 years, and 29 (11.9%) for 1–4 years. Regarding work experience in nursing, 38.0% (*n* = 81) of anesthesia nurses had more than 10 years of experience, 81 (38.0%) had been working for 5–10 years, and 18 (8.5%) for 1–3 years.

All participants were working in a major university-affiliated teaching hospital. According to the replies from anesthesiologists, most respondents (*n* = 228, 93.8%) worked in a hospital with more than 1000 beds and nearly half of them (*n* = 117, 48.2%) worked in a hospital with 1000–3000 beds. The number of surgeries performed per year varied from more than 80,000 (*n* = 42, 17.3%) to fewer than 10,000 (*n* = 9, 3.7%). The number of general anesthetic procedures performed per year also varied from less than 10,000 (*n* = 33, 13.6%) to more than 80,000 (*n* = 33, 13.6%), and most respondents (*n* = 141, 58.0%) worked in hospitals that performed 10,000–50,000 general anesthesia procedures per year. Table 1 presents a comprehensive summary of the general data of respondents.

### 3.2. Indicated Importance of ED

ED was considered to be “very important” by 357 participants (78.3%), including 195 (80.2%) anesthesiologists and 162 (76.1%) anesthesia nurses; 90 participants (evenly split between physicians and nurses) considered it to be “important” (19.7%). None of the participants answered that ED was “not important”. Only nine respondents (four anesthesiologists and five anesthesia nurses) reported that ED was “not very important” (2.0%) (Figure 1A).

Regarding the question “Do you think the management of ED is an important clinical issue?”, 141 (58.0%) anesthesiologists and 135 (63.4%) anesthesia nurses considered it was very important to intervene in ED; 99 (40.7%) anesthesiologists and 78 (36.6%) anesthesia nurses stated that it was important. Only one respondent answered that it was not important (Figure 1B).

In terms of the importance of anesthesia nurses in the management of ED, the answer “very important” (51.9% of anesthesiologists and 64.8% of anesthesia nurses) or “important” (28.4% of anesthesiologists and 19.7% of anesthesia nurses) was chosen by most of the respondents. Only three (3.7%) anesthesiologists stated that it was not important (Figure 1C).

### 3.3. Risk Factors and Intervention Measures

We analyzed responses regarding the possible risk factors and intervention measures for ED. Multiple answers were acceptable. The replies were basically consistent among anesthesiologists and anesthesia nurses. In terms of risk factors, the four most frequently chosen answers were “age,” “co-morbidities,” “pain,” and “surgery duration” (Table 2).

Regarding the intervention measures, the three most frequently chosen options were “comprehensive preoperative evaluation,” “adequate perioperative analgesia,” and “early identification” (Table 3).

### 3.4. Assessment of ED

Regarding the question “Do you routinely assess ED”, there was some inconsistency between physicians and nurses. Routine assessment of ED was more common among anesthesia nurses than anesthesiologists (25.4% vs. 8.6%, *p* = 0.006). In contrast, most anesthesiologists preferred to assess ED only in patients presenting with symptoms (46.9% vs. 26.8%, *p* = 0.01). Among the remaining respondents, 23.5% of anesthesiologists and 29.6% of anesthesia nurses indicated that they checked for ED only in patients with risk factors. Moreover, 11.1% of anesthesiologists and 7.0% of anesthesia nurses responded that they only assessed older adult patients. Meanwhile, 9.9% of anesthesiologists and 11.3% of anesthesia nurses reported that they never assessed ED (Figure 2A).

We analyzed responses regarding tools for ED assessment. Most respondents used clinical symptoms to assess ED, including 65 (80.2%) anesthesiologists and 59 (83.1%) anesthesia nurses. Of the remaining respondents who used screening tools, different opinions emerged between physicians and nurses. Anesthesiologists were highly consistent in using the Confusion Assessment Method (CAM) to screen for ED. Nevertheless, the percentage of anesthesia nurses who used the CAM was only half that of anesthesiologists (60.7% vs. 30.2%, *p* = 0.008). Outside of the CAM, the four instruments most frequently selected by anesthesia nurses were the Clinical Assessment of Confusion (CAC) (*n* = 14, 19.7%), the Bedside Confusion Scale (BCS) (*n* = 11, 15.4%), the Cognitive Test for Delirium (CTD) (*n* = 8, 11.2%), and the Clinical Global Impressions Scale Delirium (CGID) (*n* = 4, 5.6%) (Figure 2B).

### 3.5. Monitoring Depth of Anesthesia (DOA)

There were diverse responses to the question “Do you think it is necessary to monitor the DOA?”; 50.7% of anesthesia nurses reported that they considered it should be monitored routinely, while only 29.6% of anesthesiologists provided the same response (*p* = 0.008). In contrast, 60.5% of anesthesiologists indicated that monitoring the DOA was only necessary for patients with risk factors, while only 36.6% (approximately half the percentage) of anesthesia nurses provided the same response (*p* = 0.003). The remaining 6.2% of anesthesiologists and 8.5% of anesthesia nurses reported the necessity of monitoring the DOA only in older adult patients. Only six (3.9%) participants (three physicians and three nurses) said that it was not necessary to monitor the DOA (Figure 2C).

In terms of monitoring devices, most respondents reported the bispectral index (BIS; 92.6% of anesthesiologists and 95.8% of anesthesia nurses). Less frequently provided answers included “rSO2” (25.9% of anesthesiologists and 22.5% of anesthesia nurses), “EEG” (16.0% of anesthesiologists and 14.1% of anesthesia nurses), “Sedline^®^ EEG monitor” (7.4% of anesthesiologists and 11.3% of anesthesia nurses), and “TCD” (4.9% of anesthesiologists and 5.6% of anesthesia nurses) (Figure 2D).

### 3.6. Treatment for Delirium

Responses regarding the reported therapy regimes for hyperactive ED comprised treatment strategies as well as medications. Multiple answers were available.

According to the replies, there was no unified strategy or medication used. Multiple options were reported by both physicians and nurses. Responses showing significant differences between physicians and nurses were “pain management” (85.2% vs. 70.4%, *p* = 0.001), “shout to inhibit any aggressive behavior” (58.0% vs. 76.06%, *p* = 0.001), “restraint” (54.3% vs. 76.06%, *p* = 0.001), and “artery blood gas analysis” (50.62% vs. 60.56%, *p* = 0.03) (Table 4). Furthermore, several medications were reported to be used for hyperactive delirium. The top three options chosen by anesthesiologists were “dexmedetomidine” (*n* = 31, 38.3%), “propofol” (*n* = 28, 34.57%), and “intravenous analgesic” (*n* = 14,17.28%), which all showed the same degree of popularity among anesthesia nurses. The other options included “midazolam” (2.47% of anesthesiologists and 5.63% of anesthesia nurses), “haloperidol” (3.7% of anesthesiologists and 2.8% of anesthesia nurses), and “droperidol” (2.47% of anesthesiologists and 1.41% of anesthesia nurses). Only one anesthesiologist and two nurses answered “don’t know” (Table 5).

## 4. Discussion

To our knowledge, this is not only the first study investigating the clinical management of ED in China but also the first questionnaire representing both physician’s and nurses’ perceptions of ED. We herein surveyed 93 academic hospitals distributed across all provinces and municipalities in mainland China. Our response rate was 87.1% for anesthesiologists and 76.3% for nurses, suggesting that our data are as representative as possible. According to this survey, current awareness of the importance of ED was excellent among both anesthesiologists and anesthesia nurses in major teaching hospitals across China. Furthermore, the importance of anesthesia nurses in PACU during the management of ED was also well-accepted. In terms of risk factors and possible intervention measures, both anesthesiologists and anesthesia nurses were highly consistent in their responses. ED screening was performed in many hospitals, and only 9.9% of anesthesiologists and 11.3% of anesthesia nurses said that they never screened for ED. However, ED was only assessed in certain patient populations presenting symptoms or with risk factors. Even if assessed, most respondents tended to use clinical observation. Regarding the screening tools, anesthesiologists were highly consistent in using CAM. Anesthesia nurses, however, reported multiple screening tools used during their clinical practice. Routine DOA monitoring was supported more generally by anesthesia nurses, while most anesthesiologists only monitored high-risk patients. In addition, multiple treatment algorithms were indicated by both physicians and nurses.

The published ESA guideline on POD proposed the importance of POD management. Similarly, we herein reported that 98% of respondents in this survey considered ED to be “very important” and advocated prompt intervention. In terms of incidence, most participants had encountered ED of all three subtypes (hyperactive, hypoactive, and mixed) during their practice. However, the exact incidence of ED varied because of different diagnostic criteria. One recent study from China indicated that 25% of patients undergoing elective non-cardiovascular surgery developed ED in the PACU [10]. However, the data did not include the incidence of delirium subtypes. Further investigations, especially focusing on hypoactive delirium, should be performed.

Several studies have demonstrated the role of bedside nurses in delirium detection and treatment in ICUs [11,12], palliative care units [13], and general medical wards [14]. In the current survey, most anesthesiologists and anesthesia nurses agreed that nurses play an important role in the management of ED. Specifically, delirium is preventable in approximately 30–40% of patients [15]. Since nurses can identify mental status changes early, they are always in a strategic position to improve detection rates and provide necessary care, especially for at-risk patients. Therefore, we advocate an interdisciplinary team approach in the management of ED, including the participation of nurses, which will help optimize patient outcomes.

Although there has been limited research on the exact risk factors for ED, there have been plenty of studies on the risk factors for POD, which could somehow overlap; one recent study indicated that maintenance of anesthesia with inhalation anesthetic agents, malignant primary disease, American Society of Anesthesiologist Physical Status (ASA-PS) III–V, elevated serum total or direct bilirubin, and invasive surgery were independent risk factors for patients undergoing elective non-cardiovascular surgery [16]. In our survey, “co-morbidities” was the second most frequently selected answer, following higher ASA-PS. Other frequently selected answers included “advanced age,” “pain,” and “surgery duration,” which have all been demonstrated to be risk factors for POD in several previous studies [17,18,19]. Although ED is considered to be one category of POD, the direct risk factors for the two abovementioned deliriums could be slightly different. Therefore, more prospective investigations are needed to explicitly identify the risk factors for ED to subsequently guide teaching and training. Furthermore, patients with co-morbidities that influence POD incidences, such as a history of central nervous system disease, dementia, preoperative depression, and preoperative alcohol use, were not included in the abovementioned study of risk factors for ED. Further explorations are also needed that will include these patient populations.

The ESA guideline recommends routine assessment for POD in all patients starting in the recovery room; however, according to our survey, only 8.6% of anesthesiologists and 25.4% of anesthesia nurses performed assessment routinely. In contrast, most anesthesiologists preferred to only assess patients presenting with symptoms, while most anesthesia nurses preferred to only check patients with risk factors. The reasons for the difference between physicians and nurses may be explained as follows. Anesthesia nurses mainly work in the PACU, where ED mainly occurs; therefore, they are in a position to pay more attention to its occurrence. Anesthesiologists, however, will be providing medical services to other patients after sending their previous patients to the PACU and so will not have the same opportunity for ED assessment. Moreover, compared with nurses, the physicians sampled were more experienced, so they preferred to rely on clinical observation and patient symptoms. The results from this survey indicated that the awareness of routinely screening for ED before or after sending patients to the PACU needs to be improved, especially among anesthesia nurses.

Regarding the diagnosis of ED, our survey revealed that both anesthesiologists and anesthesia nurses tended to use clinical observation to diagnose ED. However, this may result in high failure rates and have low diagnostic validity [20]. In terms of screening tools, the ESA guideline recommends the Nursing Delirium Screening Scale (Nu-DESC) and CAM for delirium assessment in the PACU [1]. Furthermore, to identify hypoactive delirium, a subtype of delirium with a high incidence rate but a low diagnostic rate that is dominated by symptoms of drowsiness and inactivity, the 4As Test (4AT) and the Nu-DESC were strongly recommended [21]. According to our survey, most anesthesiologists were highly consistent in using the CAM; in contrast, most anesthesia nurses selected inappropriate instruments, including, the CAC, BCS, CTD, and CGID. Therefore, further education about using valid instruments, either the Nu-DESC (without additional training) or CAM (requiring training), to diagnose ED and about using proper instruments for different types of delirium needs to be given, especially in anesthesia nursing teams.

A significantly higher percentage of anesthesia nurses in our survey reported that they advocated routinely monitoring DOA. In contrast, higher rates of anesthesiologists indicated that they considered monitoring DOA only in patients with risk factors. The reason for the different opinions between physicians and nurses may be that more experienced anesthesiologists tend to rely on clinical observation (vital signs), while inexperienced nurses tend to rely on monitoring devices during anesthesia maintenance. In terms of specific monitoring devices, both anesthesiologists and anesthesia nurses most frequently chose BIS, possibly owing to the higher availability of suitable devices. However, BIS is not the best option to predict ED due to several limitations, such as its inaccuracy without muscle relaxant application. Therefore, the ESA guideline recommends multiple DOA monitoring devices to prevent POD, including BIS, EEG, cerebral oximetry, etc. A systematic review showed that EEG can differentiate patients with and without delirium [22]. Similarly, the potential link between intraoperative EEG patterns and ED was investigated. The occurrence of EEG burst suppression during maintenance and type of EEG emergence trajectory may predict ED [23]. Nevertheless, in our survey, only a small proportion of participants (16.0% of anesthesiologists and 14.1% of anesthesia nurses) indicated using EEG for DOA monitoring. Therefore, more general monitoring of DOA and EEG measures during general anesthesia maintenance should be advocated in clinical practice.

According to the ESA guideline, if POD occurs, immediate treatments of both causative factors and symptoms are important to reduce its duration. Regarding first-line medications, benzodiazepines, α2-agonists (dexmedetomidine), haloperidol, atypical neuroleptics, and melatonin may be beneficial; however, previous findings have been conflicting [24]. Furthermore, few studies have explored treatment strategies and medication for ED. One randomized controlled trial showed that prophylactic administration of diphenhydramine–paracetamol reduced emergence agitation after maxillofacial surgery [25]. Another recent study demonstrated that simulation training exercises may improve outcomes of ED in patients with post-traumatic stress disorder [26]. In fact, the results from our survey regarding treatment strategies for ED present great disparity, which means further investigations are needed before a consensus on such a controversial issue can be reached.

This study had several limitations. First, the subjects of our survey were all from hospitals in big cities in mainland China and our results may not reflect the much broader practice in non-academic hospitals throughout China. Second, it may seem that physicians and nurses cannot be compared because of their different educational backgrounds; however, the original motive for this survey was not to distinguish superiority and inferiority but to objectively present the points of view of these two categories of medical professionals who are closely involved in ED management.

## 5. Conclusions

In conclusion, the present survey revealed that most anesthesiologists and anesthesia nurses in China were aware of the importance of ED, as well as the importance of the anesthesia nursing team during the management of ED. Risk factors for ED need further exploration. Valid ED screening, routine monitoring of DOA, and a standardized treatment strategy should be advocated among Chinese anesthesiologists and anesthesia nurses.

## Figures and Tables

**Figure 1 brainsci-12-00989-f001:**
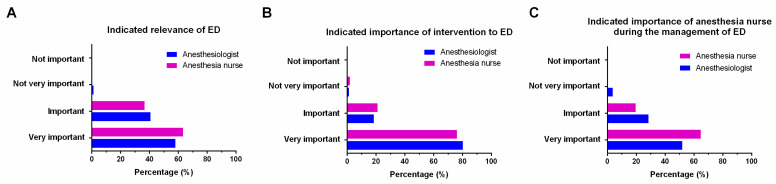
Importance of ED and ED-related issues reported by physicians and nurses, respectively. (**A**) Indicated importance of ED. (**B**) Indicated importance of intervention in ED. (**C**) Indicated importance of anesthesia nurses during the management of ED. ED, emergence delirium.

**Figure 2 brainsci-12-00989-f002:**
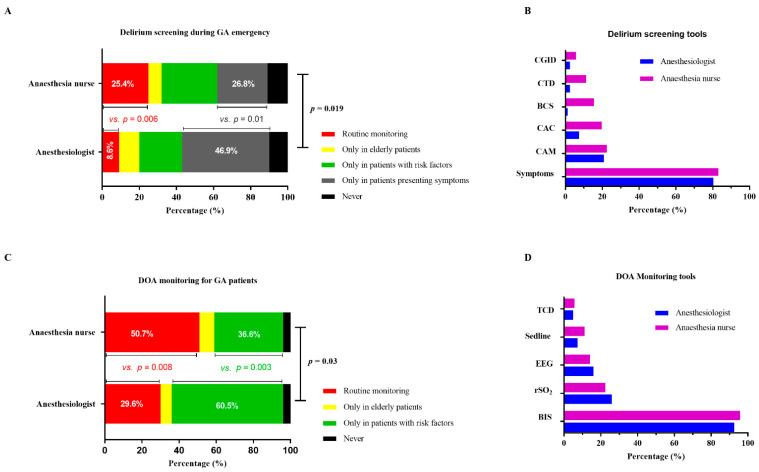
Assessment of ED and monitoring of DOA. (**A**) Delirium screening during GA emergency. (**B**) ED screening tools. (**C**) Routine monitoring of DOA for GA patients. (**D**) DOA monitoring tools. ED, emergence delirium; DOA, depth of anesthesia; GA, general anesthesia.

**Table 1 brainsci-12-00989-t001:** General data and characteristics of the respondents.

General Data	
**Profession of respondents**	
Anesthesiologist	243 (87.10%)
Nurse	213 (76.34%)
**Years in anesthesiology**	
<5 years	29 (11.93%)
5–10 years	81 (33.33%)
10–15 years	85 (34.97%)
15–20 years	27 (11.11%)
>20 years	21 (8.64%)
**Years in nursing**	
<3 years	18 (8.45%)
3–5 years	33 (15.49%)
5–7 years	51 (23.94%)
7–10 years	30 (14.08%)
>10 years	81 (38.03%)
**Beds in hospital**	
<1000	15 (6.17%)
1000–2999	117 (48.15%)
3000–4999	78 (32.10%)
5000–9999	27 (11.11%)
≥10,000	6 (2.47%)
**Surgeries/year**	
<10,000	9 (3.70%)
10,000–29,999	72 (29.63%)
30,000–49,999	42 (17.28%)
50,000–79,999	78 (32.10%)
≥80,000	42 (17.28%)
**General anesthetics/year**	
<10,000	33 (13.58%)
10,000–29,999	78 (32.10%)
30,000–49,999	63 (25.93%)
50,000–79,999	36 (14.81%)
≥80,000	33 (13.58%)

**Table 2 brainsci-12-00989-t002:** Reported risk factors associated with emergence delirium.

Risk Factors	Anesthesiologists (*n* = 243)	Anesthesia Nurses (*n* = 213)	*p*-Value
Advanced age	234 (96.30%)	195 (91.55%)	0.032
Comorbidities	213 (87.65%)	177 (83.10%)	0.17
Postoperative pain	198 (81.48%)	159 (74.65%)	0.07
Duration of surgery	186 (76.54%)	150 (70.42%)	0.14
Alcohol-related disorders	180 (74.07%)	153 (71.83%)	0.59
Catheter-related discomfort	174 (71.60%)	147 (69.01%)	0.55
Anticholinergic drug	165 (67.90%)	120 (56.34%)	0.01
Hyponatremia or hypernatraemia	156 (64.20%)	132 (61.97%)	0.62
Intraoperative bleeding	156 (64.20%)	123 (57.75%)	0.16
ASA status	153 (62.96%)	135 (63.38%)	0.93
Preoperative fluid fasting	132 (54.32%)	102 (47.89%)	0.56
Site of surgery	117 (48.15%)	90 (42.25%)	0.041

**Table 3 brainsci-12-00989-t003:** Reported prevention measures for emergence delirium.

Prevention Measures	Anesthesiologists (*n* = 243)	Anesthesia Nurses (*n* = 213)	*p*-Value
Adequate perioperative pain management	228 (93.83%)	180 (84.50%)	0.05
Preoperative risk factors evaluation	225 (92.59%)	201 (94.37%)	0.45
Promptly diagnosing POD	213 (87.65%)	186 (87.32%)	0.92
Monitor depth of anesthesia	195 (80.25%)	183 (85.92%)	0.11
No premedication with benzodiazepines	153 (62.96%)	138 (64.79%)	0.69
Fast-track surgery	135 (55.56%)	111 (52.11%)	0.10

**Table 4 brainsci-12-00989-t004:** Reported therapy strategies.

Treatment Strategies for ED	Anesthesiologists (*n* = 243)	Anesthesia Nurses (*n* = 213)	*p*-Value
Pain management	207 (85.19%)	150 (70.42%)	0.001
Sedation	168 (69.14%)	156 (73.24%)	0.34
Shout to inhibit aggressive behavior	141 (58.02%)	162 (76.06%)	0.001
Restraint	132 (54.32%)	164 (76.06%)	0.001
Artery blood gas analysis	123 (50.62%)	129 (60.56%)	0.03
Noting	0 (0%)	9 (4.23%)	0.001

**Table 5 brainsci-12-00989-t005:** Reported medications.

Medications for ED	Anesthesiologists (*n* = 243)	Anesthesia Nurses (*n* = 213)	*p*-Value
Dexamedetomidine	93 (38.27%)	66 (30.99%)	0.15
Propofol	84 (34.57%)	75 (35.21%)	0.89
Intravenous analgesic	42 (17.28%)	45 (21.13%)	0.30
Midazolam	6 (2.47%)	12 (5.63%)	0.08
Haloperidol	9 (3.70%)	6 (2.82%)	0.60
Droperidol	6 (2.47%)	3 (1.41%)	0.42
Don’t know	3 (1.23%)	6 (2.82%)	0.23

## Data Availability

All datasets used and/or analyzed during the current study are available in the Wenjuanxing repository (https://www.wjx.cn/newwjx/manage/myquestionnaires.aspx?randomt=1602323409, accessed on 30 September 2020). However, due to the language version limitation, more details about the data used in the current study are available from the corresponding author upon reasonable request.

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
