# Peer review of "A Cross-Sectional Survey on the Clinical Management of Emergence Delirium in Adults: Knowledge, Attitudes, and Practice in Mainland China"

_brainsci, 2022, doi:10.3390/brainsci12080989_

Round 1

Reviewer 1 Report

Dear author,

The presented article is a survey, and the title is “A Cross-Sectional Survey on the Clinical Management of Emergence Delirium in Adults: Knowledge, Attitude, and Practice in Mainland China" submitted to the Brain Sciences. POD is a post-operative/anesthesia complication which frequently overlooked by anesthesiologists. A nationwide survey is a good start for changing daily practices. The manuscript is well written and easy to read.  I sincerely appreciate your hard work.

 While reviewing the manuscript, I found a few points for clear understanding.

 POD is a complication in both adults and children. However, the current survey is focused on adults only. If the authors discuss the POD in the pediatric population in Discuss section, the article will be more educational for the reader.

 The preoperative assessment, intraoperative management, and postoperative care for POD are very different even in Asian countries according to a recent survey. Because China is a geographically very large country,  there would be differences even in different provinces and municipalities of China. If the authors explain some differences, it would make the article more valuable.

 ** Materials and Methods

 Page 3, Line 109:

The statistical software GraphPad Prism is not properly cited

(guide: https://www.graphpad.com/guides/prism/latest/user-guide/citing_graphpad_prism.htm )

 ** Discussion

 Page 8, Line 230:

The authors reported that 243 anesthesiologists and 213 anesthesia nurses responded and the response rate was 87.1% for anesthesiologists and 76.3% for anesthesia nurses. This means that this survey is distributed to 279 anesthesiologists and 279 anesthesia nurses. The authors also reported that 93.83% of respondents worked in a hospital with more than 1000 beds. This leads to the conclusion that only a small proportion of anesthesiologists and anesthesia nurses received and participated in the survey.

 The authors wrote that the results of the current survey are as representative as possible. However, to make it clear, the total number of anesthesiologists and anesthesia nurses of the participated hospitals and the proportion of survey participants should be mentioned.

 Page 8, Line 241:

“Routine DOA was supported…”

In the current manuscript, DOA is an abbreviation of “depth of anesthesia” ( section 3.5, Page 6 ). The sentence should be modified to “Routine DOA monitoring was supported…”

 Page 9, Line 263-267:

This sentence should be mended.

Reference 16 is about POD not ED. ED is different from POD. According to recent recommendations from expert panels, even ED is not a prerequisite for POD. ( Evered L, Silbert B, Knopman DS, et al. Recommendations for the nomenclature of cognitive change associated with anaesthesia and surgery-2018. Br J Anaesth. 2018;121(5):1005-1012. doi:10.1016/j.bja.2017.11.087 )

 Page 9, Line 314

The authors wrote that the ESA guideline recommends EEG monitoring (not BIS) to prevent POD. Reference is needed for the statement. As far as I know, ESA 2017 guideline recommends DOA, not EEG, monitoring.

 Page 10, Line 324-326:

For the first-line medications of POD, a reference is needed.

 ** Data Availability Statement

The provided data repository URL is not working.

The survey content including the questionnaire and response to each question must be made public.

 -end of the review-

Reviewer 2 Report

Introduction. It would be interesting to add information about the training physicians and nurses have in managing delirium, the existence of training programs, postgraduate training ...

This is an interesting work showing the knowledge and management level of post-surgical ED among nurses and anesthesiologists in China.

I think the methodological design is straightforward, and the results are well presented.

The discussion is well founded, and the conclusions reached by the authors are based on the results obtained

Author Response

Dear reviewer: Thank you very much for your comments. We appreciated all the hard working you have done to our manuscript. Best wishes!

Reviewer 3 Report

The introduction provide sufficient background, and include all relevant references.

The methods are adequately described.

The results are clearly presented.

The conclusions are supported by the results.

Author Response

Dear reviewer: Thanks a lot for your comments and we appreciated all you have done to our manuscript. Best wishes!

Reviewer 4 Report

This cross-sectional study investigated anesthesiologists' and nurses' knowledge and attitudes regarding emergence delirium in the acute postoperative period. Although simple in nature, this was an important study that will serve as a basis for future research on emergence delirium in the acute postoperative period.  The manuscript is well-written, with clear conclusions drawn, and a discussion of future research.

I recommend only minor comment.

I think it would be easier for the reader to understand your topic if you describe the difference from the well-known delirium in the ICU.

The survey form should be added supplementary file.

Author Response

Dear reviewer: Thank you very much for your comments.

We would like to answer your questions one by one as followed:

Point 1: You mentioned to describe a little bit the difference between ED and the well-known delirium in ICU.

Response 1: It is a great suggestion and we totally agree with you. As you mentioned, ED is quite different from the traditional delirium postoperatively because it happened much earlier. And we did emphasized the definition of the ED at the very beginning of the article. Please see the instruction part, page 2 line 54. "Emergence delirium refers to the very early onset of postoperative delirium in the immediate post-anesthesia period, before or on arrival to the recovery room".

Point 2: you mentioned the survey form should be added as the supplementary file.

Response 2: We totally agree with you and we added this part as the supplementary file of the current manuscript. 

Round 2

Reviewer 1 Report

According to the authors, three doctors and three nurses in a hospital responded. In the published surveys on POD, there are huge differences in attitudes toward ED and POD among medical staff even in the same hospital. Moreover, how the respondents were selected is vague.

Therefore, it seems unreasonable to generalize the consensus from these few responses.

Author Response

Dear reviewer, thank you again for your remarkable suggestion.  As you mentioned, three respondents of each anesthesia-related staff group (anesthesiologists and anesthesia nurses) seems not a big number, however, we did not choose the respondents randomly, instead, both the three anesthesiologists and the three anesthesia nurses we have chosen were according to their professional qualifications. To be more specific, the anesthesiologists we chose were one resident in training one attending doctor and one associate chief physician, respectively. Besides, the anesthesia nurses we chose were one nurse, one supervisor nurse and one co-chief superintendent nurse, respectively. Abovementioned professional positions were all according to Chinese medical professional position system. Therefore, the opinions from the three of doctors and nurses we chose could maximumly represent the medical workers in the same level.  

We also edited the methods and materials part, please refer to the page 3, line 115 for more details